# New, simplified versus standard photodynamic therapy (PDT) regimen for superficial and nodular basal cell carcinoma (BCC): A single-blind, non-inferiority, randomised controlled multicentre study

Eidi Christensen[1,2]*, Erik Mørk[2], Olav Andreas Foss[3], Cato Mørk[4], Susanne Kroon[5], Lars Kåre Dotterud[6], Per Helsing[7], Øystein Vatne[8†], Eirik Skogvoll[9,10], Patricia Mjønes[2,11], Ingeborg Margrethe Bachmann[12,13]

1 Department of Dermatology, Clinic of Orthopaedics, Rheumatology and Dermatology, St. Olav's Hospital, Trondheim University Hospital, Trondheim, Norway, 2 Faculty of Medicine, Department of Clinical and Molecular Medicine, Norwegian University of Science and Technology (NTNU), Trondheim, Norway, 3 Orthopaedic Research Centre, Clinic of Orthopaedics, Rheumatology and Dermatology, St. Olav's Hospital, Trondheim University Hospital, Trondheim, Norway, 4 Akershus Dermatology Centre, Lørenskog, Norway, 5 Department of Dermatology and Venerology, Stavanger University Hospital, Stavanger, Norway, 6 Lillehammer Dermatology Centre, Lillehammer, Norway, 7 Department of Dermatology, Oslo University Hospital Rikshospitalet, Oslo, Norway, 8 Department of Dermatology, Førde Central Hospital, Førde, Norway, 9 Faculty of Medicine, Department of Circulation and Medical Imaging, Norwegian University of Science and Technology, Trondheim, Norway, 10 Department of Anaesthesiology and Intensive Care Medicine, St. Olav Hospital, Trondheim University Hospital, Trondheim, Norway, 11 Department of Pathology and Medical Genetics, St. Olav's Hospital, Trondheim University Hospital, Trondheim, Norway, 12 Department of Clinical Medicine, University of Bergen, Bergen, Norway, 13 Department of Dermatology, Haukeland University Hospital, Bergen, Norway

† Deceased.
* eidi.christensen@ntnu.no

**Data Availability Statement:** The dataset generated and analyzed in the present study is not

## Abstract

### Background

Topical photodynamic therapy (PDT) is an approved and widely used treatment for low-risk basal cell carcinoma (BCC), comprising two sessions with an interval of 1 week. Simplification of the treatment course can be cost-effective, easier to organize, and cause less discomfort for the patients.

### Methods and findings

We performed an investigator-initiated, single-blind, non-inferiority, randomized controlled multicentre study with the objective of investigating whether a simpler and more flexible PDT regimen was not >10% less effective than the standard double PDT in the treatment of primary, superficial, and nodular ≤2 mm-thick BCC and evaluate the cosmetic outcome. With a non-inferiority margin of 0.1 and an expected probability complete response of 0.85, 190 tumours were required in each group. Histologically verified BCCs from seven centres in Norway were randomly assigned (1:1) to either receive a new regimen of single PDT with

publicly available because permission has not been granted from the participants or the Ethical Committee. Requests for a minimal data set may be directed to the Head of Department at Department of Clinical and Molecular Medicine, Norwegian University of Science and Technology (kontakt@ikom.ntnu.no).

**Funding:** This investigator-initiated study was funded by grants from the Liaison Committee for Education, Research, and Innovation in Central Norway (project numbers 2011/16822 and 2019/ 38881), grant from St. Olav's Hospital, Trondheim University Hospital (grant number 15/9116-128) and a grant from the Cancer Fund (Kreftfondet), St. Olav's Hospital, Trondheim University Hospital, Trondheim, Norway. The funders had no role in study design, data collection and analysis, decision to publish, or preparation of the manuscript.

**Competing interests:** The authors have declared that no competing interests exist.

one possible re-treatment of non-complete responding tumours, or the standard regimen. The primary endpoint was the number of tumours with complete response or treatment failure at 36 months of follow-up, assessed by investigators blinded to the treatment regimen. Intention-to-treat and per-protocol analyses were performed. The cosmetic outcome was recorded. The study was registered with ClinicalTrials.gov, NCT-01482104, and EudraCT, 2011-004797-28.

A total of 402 BCCs in 246 patients were included; 209 tumours assigned to the new and 193 to the standard regimen. After 36 months, there were 61 treatment failures with the new and 34 failures with the standard regimen. Complete response rate was 69.5% in the new and 81.1% in the standard treatment group. The difference was 11.6% (upper 97.5% CI 20.3), i.e. > than the non-inferiority margin of 10%. Cosmetic outcomes were excellent or good in 92% and 89% following the new and standard regimens, respectively.

## Conclusions

Single PDT with possible re-treatment of primary, superficial, and nodular $\leq$ 2-mm-thick BCC was significantly less effective than the approved standard double treatment. The cosmetic outcome was favorable and comparable between the two treatment groups.

## Introduction

Basal cell carcinoma (BCC) is the most common type of skin cancer in the adult white population, with the highest incidence in Australia (>1000/100000 person-years) [1]. It poses a significant health issue due to the considerable patient morbidity and a substantial financial burden on healthcare systems [2,3]. Though the tumour usually grows slowly and rarely metastasises, it can cause extensive tissue destruction if inadequately treated or untreated [4]. BCCs are commonly classified into high and low risk, indicating the possibility of recurrence after treatment. The low-risk group includes primary, superficial, and nodular tumours of smaller size located outside the neck and mid-face zone [2]. Although surgical excision is widely used to treat BCC, photodynamic therapy (PDT) is an attractive modality for treatment of low-risk tumours, owing to good compliance, a high response rate, short healing time, few side effects, and favourable cosmetic outcomes [5]. PDT is not recommended for high-risk tumours. Commonly, PDT of BCC involves the use of 5-aminolevulinic acid (ALA) or its methyl ester metylaminolevulinate (MAL) as precursors to potent photosensitizers administered in gel or cream formulation, which causes selective accumulation of photoactive porphyrins in the tumour cells 3 h following application [6]. The porphyrins generate reactive oxygen species on illumination under red light causing cell death by necrosis and apoptosis. PDT has been extensively used as a treatment modality for non-melanoma skin cancer in the last 20 years and is approved for low-risk, superficial and small, nodular $\leq$ 2-mm-thick BCC administered in two sessions at an interval of 1 week [5,7]. The practice of double PDT emerges from results of early, open-label clinical studies that report increased PDT efficacy with the use of repeated treatment [8,9]. However, some single PDT studies are reported to achieve complete response rates of 64%-84% at 3–6 years following treatment [7,10–13]. This indicates that several cases of BCC require only one treatment, as they may be overtreated with the current standard PDT regimen. Overtreatment constitutes a large healthcare expenditure [14], and the current PDT practice may also be less cost-effective than other non-invasive treatment options [3,15].

Randomised controlled studies are needed to compare the outcomes after single versus repeated PDT. In addition, with current practice, two time-consuming hospital visits are required, which could increase patient burden. Therefore, a more flexible PDT regimen should be explored to simplify treatment.

The aim of our study was to investigate whether a simplified and more flexible PDT regimen consisting of a single PDT session with the option of one re-treatment of those BCCs with incomplete response was not >10% less effective than the approved standard double PDT regimen in the treatment of superficial and nodular ≤2-mm-thick BCC. We also aimed to evaluate the cosmetic outcome and in addition explore prognostic factors such as the patient's sex and age and tumour location, size, clinical, and histological subtypes, and thickness that may contribute to treatment failure in the groups.

## Materials and methods

### Study design

This was a single-blind, non-inferior, randomised, controlled multicentre study.

### Participants, eligibility criteria and settings

Patients recruited from the participating study sites, above 18 years of age, with one or more histologically confirmed BCC clinically assessed as non-pigmented superficial or nodular subtype, and ≤2.0 mm thick, were assessed for eligibility. Patient exclusion criteria were: child-bearing potential, Gorlin syndrome, porphyria, xeroderma pigmentosum, history of arsenic exposure or known allergy to MAL, concomitant treatment with immunosuppressive medication, or physical or mental conditions that would prevent them from attending the follow-up visits. Tumours were excluded if located on the neck or within the mid-face area, having undergone prior treatment, or had the longest diameter >15 mm on the face or scalp, >30 mm on the trunk, and >20 mm on the limbs.

The study was conducted in hospital settings, including four different university hospitals, one district general hospital, and two private dermatology clinics (Table 1). The investigators were all dermatologists and members of the Norwegian PDT group with each 15–20 years of experience in PDT.

The study was performed in compliance with the Declaration of Helsinki and the International Conference on Harmonization Guidelines for Good Clinical Practice. All patients provided written informed consent before study entry. The study protocol and consent documents were approved by the Regional Committee for Medical and Health Research Ethics (REC) Midt (reference number 2011/2048) and the Norwegian Medicines Agency (www. legemiddelverket.no, reference number 12/00273-10). The study was registered at Clinical-Trials.gov (number NCT-01482104) with EudraCT, 2011-004797-28. The Clinical Research Unit Central Norway of the Norwegian University of Science and Technology (NTNU) was responsible for randomisation and monitoring.

### Clinical and histological examinations

The clinical examination defined tumour sizes as the mean value of the maximum length and width. BCCs were clinically classified as superficial or nodular subtypes based on recognized clinical features after inspection and palpation of the tumours [16,17]. Each tumour was marked on a body chart, and most were photographed before treatment for reliable identification of the treatment area at follow-up visits.

**Table 1. Baseline distribution of patients and basal cell carcinoma characteristics in the two randomised groups.**

| Characteristics | Treatment Regimen | |
| --- | --- | --- |
| | New | Standard |
| Sex, n (%) | | |
| Male | 65 (55.6) | 63 (48.8) |
| Female | 52 (44.4) | 66 (51.2) |
| Missing n (%) | 0 (0) | 0 (0) |
| Age (years), mean (min-max.) | 66 (26–92) | 66 (37–91) |
| Missing n (%) | 0 (0) | 0 (0) |
| Previous BCC in medical history, n (%) | 108 (51.7) | 96 (49.7) |
| Missing n (%) | 6 (2.9) | 2 (1.0) |
| Fitzpatrick skin type, median (%) | | |
| I | 24 (11.5) | 25 (13) |
| II | 92 (44) | 74 (38.5) |
| III | 92 (44) | 93 (48.4) |
| IV | 1 (0.5) | 0 |
| Missing n (%) | 1 (0.5) | 1 (0.5) |
| Tumour location, n (%) | | |
| Head/ neck | 27 (12.9) | 20 (10.4) |
| Trunk | 144 (68.9) | 129 (66.8) |
| Extremities | 38 (18.2) | 44 (22.8) |
| Missing n (%) | 0 (0) | 0 (0) |
| Tumour size, (mm.) | | |
| Mean (SD, min-max) | 11.1 (4.4, 5.0–30.0) | 11.4 (4.4, 5.0–26.5) |
| Median (25%-75%, percentile) | 10.0 (8.0–13.0) | 10.0 (8.3–15.0) |
| Missing n (%) | 1 (0.5) | 0 (0) |
| Clinical tumour thickness (mm.) | | |
| Mean (SD, min-max) | 1.0 (0.5, 0.1–2.0) | 1.0 (0.6, 0.1–2.0) |
| Median (25%-75%, percentile) | 1.0 (0.5–1.0) | 1.0 (0.5–1.1) |
| Missing n (%) | 2 (1.0) | 1 (0.5) |
| Clinical tumour subtype, n (%) | | |
| Superficial | 150 (71.8) | 149 (77.2) |
| Nodular | 56 (27.3) | 42 (21.8) |
| Missing n (%) | 2 (1.0) | 2 (1.0) |
| Histological tumour thickness, (mm,) | | |
| Mean (SD, min-max) | 0.9 (0.8, 0.2–4.6) | 0.9 (0.7, 0.2–3.3) |
| Median (25%-75%, percentile) | 0.6 (0.3–1.2) | 0.5 (0.3–1.3) |
| Missing n (%) | 18 (8.6) | 18 (9.3) |
| Histological tumours subtype, n (%) | | |
| Superficial | 126 (60.3) | 121 (62.7) |
| Nodular | 46 (22.0) | 45 (23.3) |
| Aggressive | 29 (13.9) | 24 (12.4) |
| Missing n (%) | 8 (3.8) | 3 (1.6) |
| Study sites, n (%) | | |
| Akershus Dermatology Centre | 52 (24.9) | 50 (25.9) |
| Districts General Hospital in Førde | 10 (4.8) | 2 (1.0) |
| Haukeland University Hospital | 33 (15.8) | 28 (14.5) |
| Lillehammer Dermatology Centre | 21 (10.0) | 21 (10.9) |
| Oslo University Hospital | 38 (18.2) | 37 (19.2) |
| Stavanger University Hospital | 23 (11.0) | 25 (13.0) |
| St. Olav's University Hospital | 32 (15.3) | 30 (15.5) |
| Missing n (%) | 0 (0) | 0 (0) |

The investigating dermatologists obtained tissue samples from each tumour using a disposable 3 mm or 4 mm biopsy punch from the tumour area which clinically was evaluated as the thickest. Pathologists at the pathology laboratories affiliated with the study sites performed the initial histological examination to confirm the BCC diagnosis. After PDT and subsequent follow-ups, a second histological investigation of the biopsy samples was performed to assess

tumour subtype and thickness. The original biopsy blocks were transferred to the Cellular and Molecular Imaging Core Facility (CMIC), NTNU for preparation before examination by a pathologist at St. Olav's University Hospital with an extensive experience in evaluating BCC. The tumours were classified into three subtypes: superficial, nodular, or aggressive (morphoea-form, infiltrative, and micronodular) growth types [18]. If presenting a mixed-growth pattern, these were classified according to the most aggressive component. Tumour thickness was measured from below the stratum corneum to the deepest point of invasion using an oculometer (1 mm squares) or/and an ocular micrometre to a precision of 0.1 mm.

### Interventions

Before PDT, the BCC surface and 5 mm of surrounding clinical non-involved skin were prepared using a sharp disposable curette by which the scraping was performed in a checked pattern to remove any crust and superficially hard keratotic tissue [19]. If the clinical examination identified the need for further thickness reduction, selected tumours were also debulked. A similar pre-treatment procedure was repeated before the second PDT.

PDT was performed with MAL (Metvix®) 160 mg/g cream Galderma, France), which was applied to the prepared treatment area in an approximate 1-mm-thick layer. Thereafter, the area was covered with a plastic film and a lightproof occlusive bandage. The cream was left for 3 h before being removed and the treatment area was exposed to light-emitting diodes (Aktilite®) with a peak wavelength of 630 nm, fluence rate of 70 x 100 mW/cm$^2$ and exposure typically for 7–9 minutes giving a total light dose of about 37 J/cm$^2$. Treatment was repeated after 1 week with the standard regimen and after 3 months for tumours with clinical treatment failure with the new regimen.

### Outcomes

Treatment outcomes were evaluated at 3-, 12- and 36-months following PDT (Table 2).

The treatment outcome "complete response" was defined as clinical clearance of tumour in the treatment areas, including a dermatoscopic investigation at a 36-month follow-up. When in doubt, a biopsy was taken and where histological examination showed no remnants of BCC, the treatment result was recorded as a complete response. The outcome "treatment failure" was defined as clinical suspicion of failure with a following histological confirmation of remnant BCC in the treatment area during follow-up. The exception was BCC treated once in accordance with the new regimen that, after clinical suspicion of failure at a 3-month follow-up, underwent a second PDT without prior biopsy (Table 2). Tumours with observed treatment failure were terminated and further treated outside the study at the discretion of the investigators.

**Table 2. Methods used for evaluation of photodynamic therapy (PDT) outcome at follow-up.**

| Treatment Regimen | Follow-up | | | |
|---|---|---|---|---|
| | 3-month | 3-month after re-PDT of tumours with initial 3-month clinical treatment failure | 12-month | 36-month |
| **New** | Clinical and cosmetic assessment. | Clinical and cosmetic assessment. Biopsy of clinically assessed failures. | Clinical and cosmetic assessment. Biopsy of clinically assessed failures. | Clinical, dermatoscopic and cosmetic assessment. Biopsy of clinical and/or dermatoscopic assessed failures. |
| **Standard** | Clinical and cosmetic assessment. Biopsy of clinically assessed failures. | NA | Clinical and cosmetic assessment. Biopsy of clinically assessed failures. | Clinical, dermatoscopic and cosmetic assessment. Biopsy of clinically and/or dermatoscopic assessed failures |

The cosmetic outcome was assessed at 3-, 12-, and 36-month follow-up by visual inspection and palpation of the treatment areas. For those tumours treated twice using the new regimen, the initial assessment occurred 3 months after the second PDT. Cosmetic results from areas with treatment failure were not included in the analyses. The results were categorised on a four-point ordinal scale as either excellent (absence of any stigmata other than scar formation after diagnostic punch biopsy), good (slight presence of fibrosis, atrophy, or change in pigmentation), fair (moderate presence of fibrosis, atrophy, or change in pigmentation), or poor (marked presence of fibrosis, atrophy, or change in pigmentation).

Any adverse events (AEs) that occurred in the period from treatment to the 3-month follow-up were reported and described by their duration, severity, relationship to treatment and according to the need of other specific therapy. Serious adverse event (SAE) were to be reported according to specified procedures whether they were considered related to study treatment or not. Local reactions, such as erythema, pain, and weeping, were regarded as conceivable events and reported as number of days present. AEs could be reported spontaneously by the patient or through open (non-leading) questioning.

## Sample size calculation

Sample size was determined by StatXact version 9.0 (Cytel Software Corp, Waltham, USA), based on anticipated complete response probability of 0.85 obtained from early publications [6,7] and a non-inferiority margin of 10%; thus, aiming to demonstrate that the new regimen was not >0.1 inferior to the standard regimen. With a significance level at 0.05 and power at 0.80, each group required 190 tumours. Because multiple tumours were randomised within patients, no adjustments for patient identity were made.

## Randomisation

BCCs were randomised to receive the new or the standard treatment by use of a web-based system developed and administered by the Faculty of Medicine and Health Sciences, NTNU. Block randomisation was done by center where both the order of block sizes and allocation sequence of each block were generated consecutively by the system. An administrator initiated the randomisation system and the assignment was sent by email to the appointed study investigator who carried out the treatment.

To ensure unpredictability of the random allocation in patients with multiple BCCs, tumours were numbered consecutively, and recorded on the body map included in the case report forms (CRFs) before randomisation. The distance between BCCs had to be clinically ≥30 mm apart to be regarded as two individual tumours. The numbering started on the front side of the patient's body and from top to bottom. If two tumours were located on the same horizontal line, the numbering first followed the tumour located furthest to the right side of the patient's body. The corresponding system was then applied to the patient's back. The first tumour was randomised to one of the two treatment regimens and the second was allocated to the other regimen. A third tumour was randomised to one of the two treatment regimens and a fourth allocated to the other regimen and so on. The new PDT regimen included one single PDT with one possible re-treatment of clinically non-complete responding tumours at the initial 3-month follow-up. The standard treatment regimen included two PDT treatments at an interval of 1 week. To reduce treatment bias, the investigators performed the tumour preparation before randomisation.

## Blinding

Both treatment and cosmetic outcomes were evaluated by dermatologists working at the study centres blinded to the treatment regimen.

## Statistical analyses

The final analysis was performed by StatExact version 10.0 (Cytel Software Corp.).

We used an exact non-inferiority test with a margin of 0.1 and corresponding one-sided 97.5% confidence intervals (CI) since one-sided CIs are customary in non-inferiority studies.

We considered six different scenarios to reflect various assumptions about the loss to follow-up. The first scenario represents "per-protocol" analyses, including only tumours with 36-month follow-up results. The next two scenarios represent "intention-to-treat": a "best-case" scenario in which tumours lost to follow-up were categorised as in complete response and a "worst-case" scenario in which missing results were categorised as treatment failures. Thereafter, three similar analyses were performed but restricted to tumours histopathologically evaluated to have been suitable for PDT treatment. Thus, histological aggressive subtypes and/or tumours thicker than 2 mm were excluded. Stacked bar graphs were used to report cosmetic outcomes. Conceivable AEs were reported as median (25–75) percentiles. Descriptive statistics were used when reporting patients and tumour characteristics. Box plots were used when reporting patient age, BCC size, and thickness of tumours with complete response.

## Results

Between June 2012 and April 2014, a total of 402 BCCs from 246 patients were included and randomised, 209 tumours to the new regimen, and 193 tumours to the standard regimen without any crossovers between groups during the treatment period. One tumour was treated in each of 163 patients, two tumours in each of 45 patients, three tumours in each of 18 patients, four tumours in each of 9 patients, five tumours in each of 8 patients, six tumours in each of 2 patients, and seven tumours in one patient. Data on patient demographics, tumour characteristics, and treatment centres are presented in Table 1. The distributions were similar in the two treatment groups. Fig 1 presents the flow diagram of tumours. In the new regimen group, 29 cases (14%) were evaluated clinically at the initial 3-month follow-up as treatment failures and were treated with a second PDT. Three patients asked for an unscheduled 24-month follow-up owing to suspected tumour relapse. Data on the endpoint for treatment response were not available for 22 tumours (9 in the new regimen and 13 in the standard regimen), owing to circumstances including treatment deviations (i.e., tumour receiving treatment other than PDT), withdrawal of consent to participate in the study, patients not attending follow-ups, and patient death.

At the 36-month follow-up, we observed 61 treatment failures by the new regimen and 34 failures by the standard regimen. Table 3 shows the results of the treatment effect at the 36-month follow-up. Complete response rate was 81.1% in the standard treatment group and 69.5% in the new treatment group, with a difference of 11.6% (upper 97.5% CI 20.3, p = 0.64), exceeding the non-inferiority margin of 10%.

The difference in the best-case scenario was 11.6 (upper 97.5% CI 19.8, p = 0.65), and in the worst-case scenario was 9.1% (upper 97.5% CI 18.1, p = 0.43), both larger than the non-inferiority margin. The 36-month complete tumour response for tumours that were histopathological suitable for PDT (subtype and thickness) is presented in Table 4, of which both exceeded the non-inferiority margin. The cosmetic outcome at the 36-month follow-up was recorded as excellent or good in 128 of 139 (92%) of the evaluated treatment areas by the new regimen and in 132 of 146 (89%) areas by the standard regimen. More detailed information on the cosmetic outcome is given in Fig 2.

The mean (min -max) number of days with erythema, weeping, and pain after PDT were 7 (5–14), 1 (0–3), and 0 (0–1) for the new regimen, respectively, and 7 (7–14), 0 (0–3), and 0 (0–

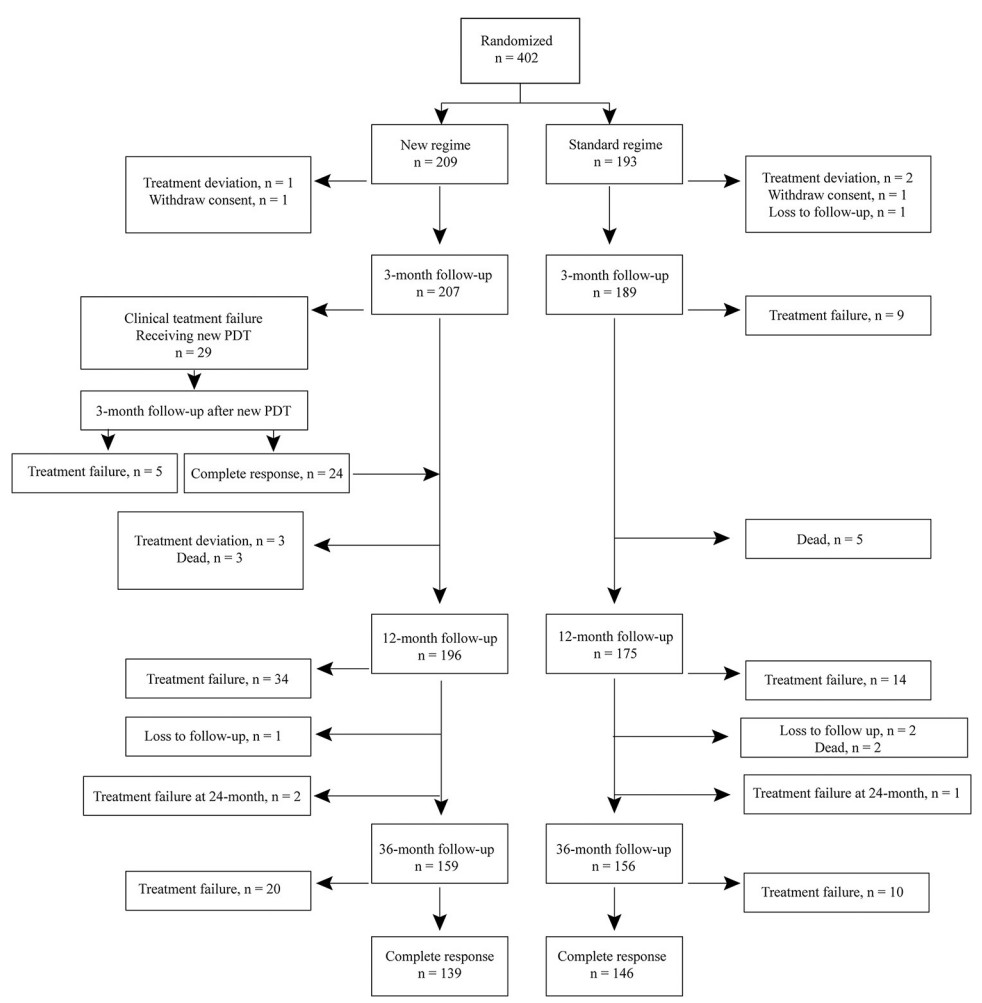

**Fig 1. Flow diagram of basal cell carcinoma.**

**Table 3. Outcomes after photodynamic therapy of basal cell carcinoma at 36-month follow-up.**

| Analysis scenario | | Treatment regimen | | | | Difference (% of non-failure) | 97.5% CI Upper | P value |
|---|---|---|---|---|---|---|---|---|
| | | Standard | | New | | | | |
| | | Non-failure | Failure | Non-failure | Failure | | | |
| Per protocol | Number | 146 | 34 | 139 | 61 | 11.6 | 20.3 | 0.64 |
| | % | 81.1 | 18.9 | 69.5 | 30.5 | | | |
| ITT, best case[a] | Number | 159 | 34 | 148 | 61 | 11.6 | 19.8 | 0.65 |
| | % | 82.4 | 17.6 | 70.8 | 29.2 | | | |
| ITT, worst case[b] | Number | 146 | 47 | 139 | 70 | 9.1 | 18.1 | 0.43 |
| | % | 75.6 | 24.4 | 66.5 | 33.5 | | | |

ITT, intention-to-treat.

[a] outcomes in which tumours lost to follow-up were categorised as in complete response.

[b] outcomes in which tumours lost to follow-up were categorised as treatment failures.

**Table 4. Outcomes after photodynamic therapy of basal cell carcinoma histologically suitable for treatment at 36-month follow-up.**

| Analysis scenario | | Treatment regimen | | | | Difference (% of non-failure) | 97.5% CI Upper | P value |
|---|---|---|---|---|---|---|---|---|
| | | Standard | | New | | | | |
| | | Non-failure | Failure | Non-failure | Failure | | | |
| Per protocol | Number | 107 | 22 | 103 | 37 | 9.3 | 19.2 | 0.47 |
| | % | 82.9 | 17.1 | 73.6 | 26.4 | | | |
| ITT, best case[a] | Number | 116 | 22 | 109 | 37 | 9.4 | 18.9 | 0.47 |
| | % | 84.1 | 15.9 | 74.7 | 25.3 | | | |
| ITT, worst case[b] | Number | 107 | 31 | 103 | 43 | 7.0 | 17.2 | 0.29 |
| | % | 77.5 | 22.5 | 70.5 | 29.5 | | | |

ITT, intention-to-treat.

[a] outcomes in which tumours lost to follow-up were categorised as in complete response.

[b] outcomes in which tumours lost to follow-up were categorised as treatment failures.

3) for the standard regimen, respectively. One case of SAE was reported as an episode of fall in the home, which led to hospitalization for 3 days.

Table 5 shows tumours with complete responses related to sex, tumour location, and clinical and histological evaluation of tumour subtypes. There were minor differences in these categorical data between the two regimens. Fig 3 illustrates the tumours with complete responses related to the patient's age and tumour size, and Fig 4 shows the relationship between clinical and histological evaluation of tumour thickness of the continuous data. There were minor differences in responses between the two treatment regimens.

## Discussion

### Main findings

The main finding in this study is that a simplified regimen for PDT of BCC was inferior to the standard regimen. Both the intention-to-treat and per-protocol analyses demonstrated that complete response rates after treatment with the new regimen could be more than 10% inferior

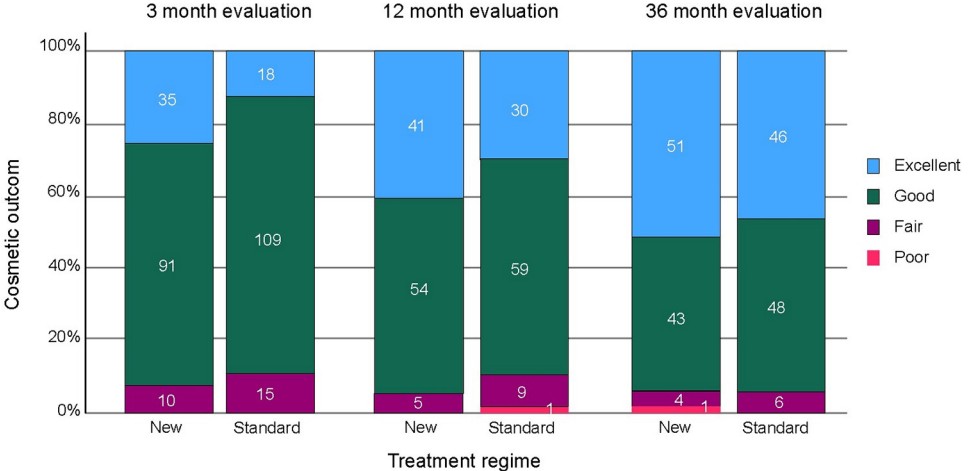

**Fig 2. Cosmetic outcomes presented as bars with percentages and numbers within each category.**

**Table 5. Patient sex, tumour location and clinical and histological subtypes of basal cell carcinoma with complete response after photodynamic therapy at 36-month follow-up.**

| | | Treatment regimen | | | |
| | | New | | Standard | |
| | | Number | % | Number | % |
|---|---|---|---|---|---|
| Gender | Male | 81 | 58.3 | 80 | 54.8 |
| | Female | 58 | 41.7 | 66 | 45.2 |
| Location | Head-neck | 16 | 11.5 | 16 | 11.0 |
| | Trunk | 96 | 69.1 | 99 | 67.8 |
| | Extremities | 27 | 19.4 | 31 | 21.2 |
| Subtypes, Clinical evaluation | Superficial | 99 | 72.3 | 113 | 78.5 |
| | Nodular | 38 | 27.7 | 31 | 21.5 |
| Subtypes, Histological evaluation | Superficial | 90 | 66.7 | 96 | 66.2 |
| | Nodular | 31 | 23.0 | 34 | 23.4 |
| | Aggressive | 14 | 10.4 | 15 | 10.3 |

compared with the standard regimen, even though 14% of tumours randomised to the new regimen had been treated twice. The inferiority margin of a maximum of 10% difference in treatment failure between the two regimens was considered acceptable because it was expected to reduce patient burden, be practically easier to organise and be more cost-effective than the conventional double treatment. The new regimen should also provide a success rate comparable with results from other minimally invasive techniques, such as cryosurgery and imiquimod [20,21]. Additionally, the results showed no clinically important difference in the cosmetic outcome between the two regimen groups. The conceivable AEs were comparable between the two groups, and no suspected unexpected SAEs occurred.

Tumour subtype and thickness were clinically evaluated to reflect common daily practice. Several BCCs are treated without prior histopathological examination [22], and in our experience, if a biopsy is taken, a description of tumour subtype and/or thickness is irregularly included in the histology report. However, it can be argued that the use of clinical assessment for selecting BCCs suitable for PDT does not exclude the presence of histologically aggressive and thick tumours. Consequently, a sub analyses was made in which only histologically observed superficial and nodular BCCs not exceeding 2 mm thickness were included. Even with such an approach, the efficacy outcomes of the new, simplified treatment regimen exceeded the 10% non-inferiority margin.

## Comparisons with other studies

The practice of two MAL-PDT sessions for BCC has been recommended for about two decades without being properly tested. To the best of our knowledge, this is the first randomised controlled study comparing a simplified regimen consisting of a single PDT session with the possibility of one re-treatment, with standard two treatments.

Complete response rates for BCC after different treatment methods depend on the length of follow-up time [23]. After PDT, most recurrences present within 3 years [7,20]. We achieved a high complete response rate of > 80% for tumours treated with two standard PDT. This is a satisfactory result compared to those from several other studies with long-term follow-up after PDT of BCC [5,24] and is superior to the outcome of a recent large randomised controlled study comparing MAL-PDT with other minimal invasive treatment methods in superficial BCC with a 3-year complete response rate of 58% for PDT [25]. Even though various patient- and BCC-related characteristics are reported to be associated with PDT failure

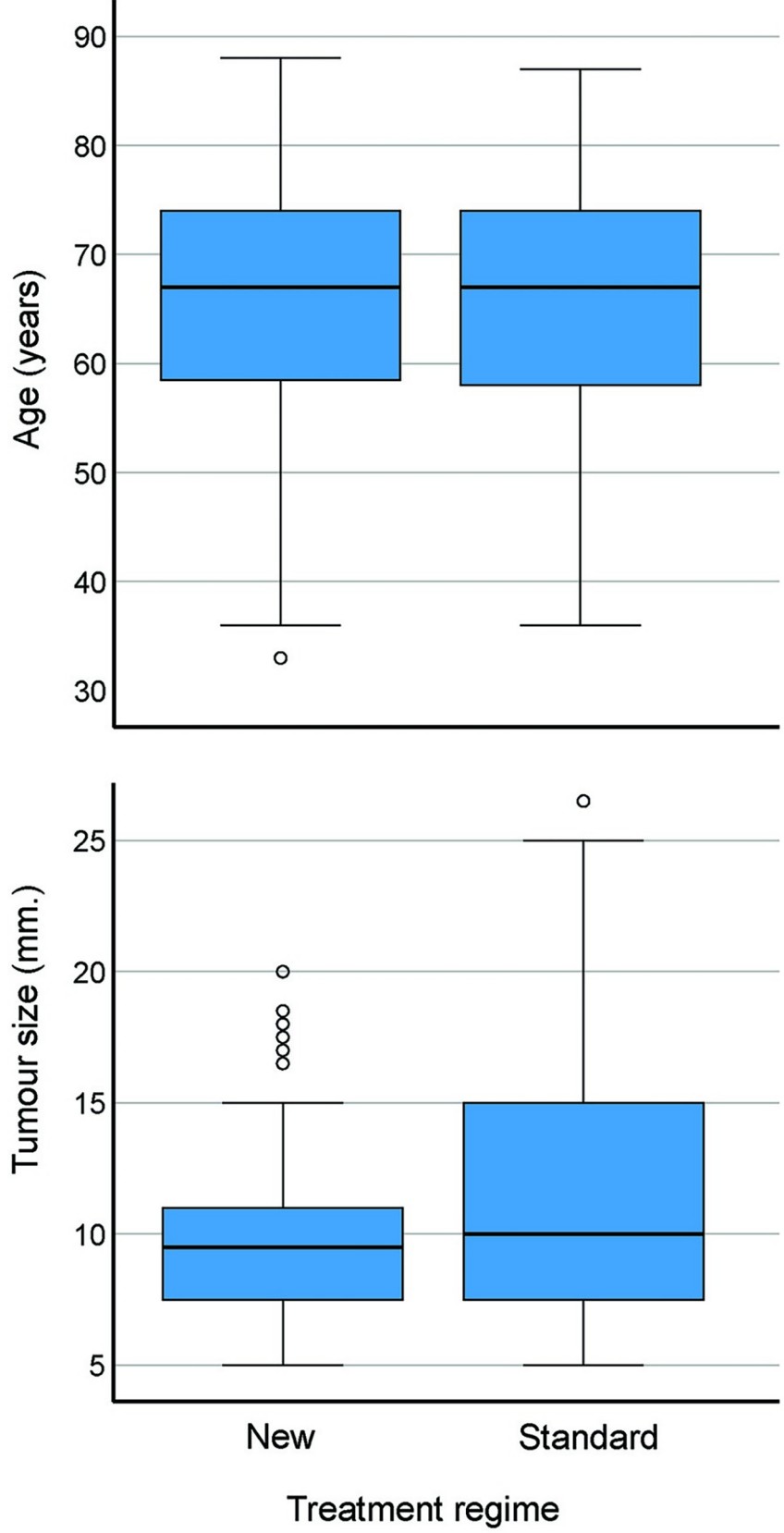

**Fig 3. Patient's age and tumour size in basal cell carcinomas with complete response at 36-month follow-up.**

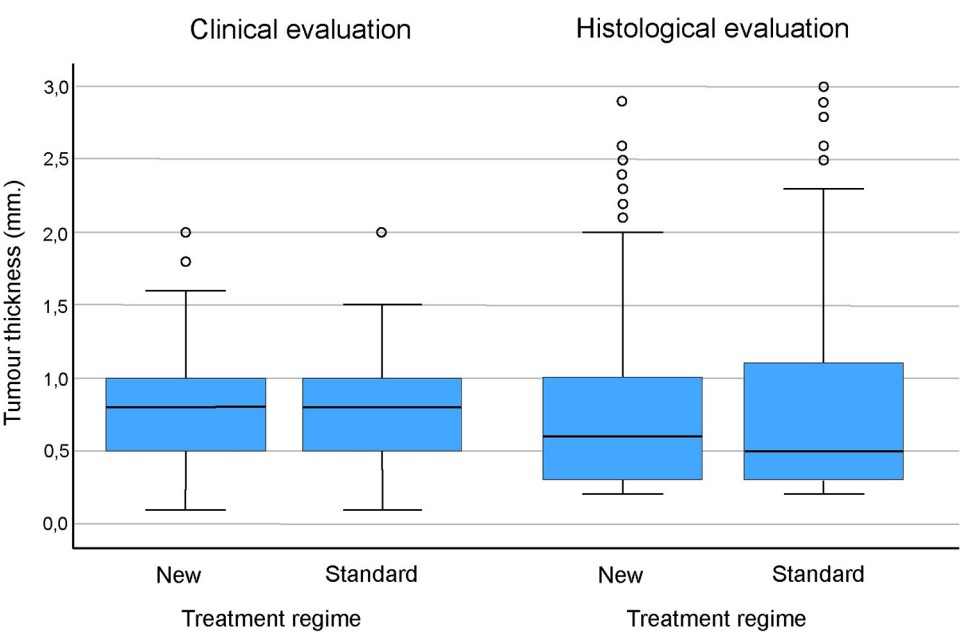

**Fig 4. Clinical and histological tumour thickness in basal cell carcinomas with complete response at 36-month follow-up.**

[26], we found no significant differences of such characteristics between the groups for BCC with complete response. However, differences in study design and execution can make a direct comparison between results difficult.

## Strengths and limitations

The strengths of this investigator-initiated study include a proper study design, a large sample size, few tumours (patients) lost to follow-up, and the use of combined resources across hospitals and private dermatology centres, which promotes generalizability of the outcomes. However, the results do not apply to all BCC but to tumours that met the study eligibility criteria and from a fair-skinned population Limitations also include that all BCCs were histologically verified before inclusion, which affects the generalizability of the results and the use of punch biopsies for histological examination since they only offer information from a small, selected area of the tumours [27]. Reporting of conceivable adverse events may have been incomplete due to the patient's different ability to recall symptoms over a 3-month period. Different dermatologists may have been involved in the assessment of treatment areas at the various centres during the study period, and this may have led to a less uniform assessment of outcomes. However, the practise was carried out in accordance with common clinical practise and may thus increase the generalizability of the results.

## Other approaches to optimize the standard PDT regimen

Although the double PDT practice is well established, optimization of the standard protocol is being pursued. Recently, the treatment efficacy at 60 months after a single PDT visit with two treatments on the same day was reported to be 80.6% [28]. Among other attempts to challenge the established PDT practice of two treatments one week apart are studies that have investigated the outcome of a single PDT of superficial BCC using fractionated irradiation protocols with one or more light fractions given on the same treatment day. These studies have shown

promising short-term complete response rates of up to 80%–95% [15,29]. However, a recent study on the long-term efficacy of fractionated ALA-PDT versus conventional double MAL-PDT showed fractionated PDT to be less effective [30].

## Conclusions

We conclude that a single session of PDT, including optional re-treatment, for primary, superficial, and nodular ≤2-mm-thick BCC was significantly inferior to the two standard treatments. Two sessions of PDT are recommended for low-risk BCC. The cosmetic outcome was highly favourable and comparable in the two groups.

## Supporting information

**S1 Checklist. Reporting checklist for randomised trial.**
(DOCX)

**S2 Checklist. SPIRIT 2013 checklist: Recommended items to address in a clinical trial protocol and related documents\*.**
(DOC)

**S1 File.**
(DOCX)

**S2 File.**
(PDF)

## Acknowledgments

First, the authors thank the patients who participate in this study. We also thank the nursing staff, the pathologists, and the clinical managers of the participating hospitals for their contributions. Further, we thank the Cellular and Molecular Imaging Core Facility (CMIC) for preparing the biopsy samples.

## Author Contributions

**Conceptualization:** Eidi Christensen, Cato Mørk, Susanne Kroon, Lars Kåre Dotterud, Per Helsing, Øystein Vatne, Eirik Skogvoll, Ingeborg Margrethe Bachmann.

**Data curation:** Eidi Christensen, Cato Mørk, Susanne Kroon, Lars Kåre Dotterud, Per Helsing, Øystein Vatne, Ingeborg Margrethe Bachmann.

**Formal analysis:** Eidi Christensen, Erik Mørk, Olav Andreas Foss, Eirik Skogvoll.

**Funding acquisition:** Eidi Christensen, Cato Mørk.

**Investigation:** Eidi Christensen, Cato Mørk, Susanne Kroon, Lars Kåre Dotterud, Per Helsing, Øystein Vatne, Patricia Mjønes, Ingeborg Margrethe Bachmann.

**Methodology:** Eidi Christensen, Cato Mørk, Susanne Kroon, Lars Kåre Dotterud, Per Helsing, Øystein Vatne, Eirik Skogvoll, Ingeborg Margrethe Bachmann.

**Project administration:** Eidi Christensen.

**Resources:** Eidi Christensen, Cato Mørk, Susanne Kroon, Lars Kåre Dotterud, Per Helsing, Øystein Vatne, Ingeborg Margrethe Bachmann.

**Supervision:** Eidi Christensen, Olav Andreas Foss, Eirik Skogvoll, Patricia Mjønes, Ingeborg Margrethe Bachmann.

**Validation:** Eidi Christensen, Erik Mørk, Olav Andreas Foss, Eirik Skogvoll, Ingeborg Margrethe Bachmann.

**Visualization:** Eidi Christensen, Erik Mørk, Olav Andreas Foss.

**Writing – original draft:** Eidi Christensen, Erik Mørk.

**Writing – review & editing:** Eidi Christensen, Erik Mørk, Olav Andreas Foss, Cato Mørk, Susanne Kroon, Lars Kåre Dotterud, Per Helsing, Øystein Vatne, Eirik Skogvoll, Patricia Mjønes, Ingeborg Margrethe Bachmann.

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
