## [Decision Letter · Decision Letter 0]

20 Nov 2023

PONE-D-23-27160New, simplified versus standard photodynamic therapy (PDT) regimen for superficial and nodular basal cell carcinoma (BCC): A single blind, non-inferiority, randomised controlled multicentre study.PLOS ONE

Dear Dr. Christensen,

Thank you for submitting your manuscript to PLOS ONE. After careful consideration, we feel that it has merit but does not fully meet PLOS ONE’s publication criteria as it currently stands. Therefore, we invite you to submit a revised version of the manuscript that addresses the points raised during the review process.

We look forward to receiving your revised manuscript.

Kind regards,

Payodun Koralage Buddhika Mahesh

Academic Editor

PLOS ONE

Reviewers' comments:

Reviewer's Responses to Questions

**Comments to the Author**

1. Is the manuscript technically sound, and do the data support the conclusions?

Reviewer #1: Yes

Reviewer #2: Yes

Reviewer #3: Partly

Reviewer #4: Yes

Reviewer #5: Yes

2. Has the statistical analysis been performed appropriately and rigorously? 

Reviewer #1: Yes

Reviewer #2: I Don't Know

Reviewer #3: Yes

Reviewer #4: Yes

Reviewer #5: N/A

3. Have the authors made all data underlying the findings in their manuscript fully available?

Reviewer #1: No

Reviewer #2: No

Reviewer #3: Yes

Reviewer #4: Yes

Reviewer #5: Yes

4. Is the manuscript presented in an intelligible fashion and written in standard English?

Reviewer #1: Yes

Reviewer #2: Yes

Reviewer #3: Yes

Reviewer #4: Yes

Reviewer #5: Yes

5. Review Comments to the Author

Reviewer #1: This presents technically sound research and the data supports the conclusions. A sub analysis between study sites can be included see any site specific variations, if any. Seems bit late in publishing as the the study was done 2012 -2014.

Reviewer #2: - Mention the objectives of the study in the abstract as well.

- Some details of the design are not clearly stated, i.e. who generated the random allocation sequence, who enrolled participants, and who assigned participants to

Interventions.

- For binary outcomes, presentation of both absolute and relative effect sizes is recommended (Table five)

- Sources of potential Bias need to be addressed in the discussion section.

Reviewer #3: Congratulations to the authors for this important study.

My suggestions to further improve the manuscript are as follows:

Abstract -

Methods: sample size calculation details need to be given.

Prognostic factors assessed for cosmetic outcomes, not mentioned.

Results: The range of the 97.5% CI needs to be given for informed decision making.

Conclusion: Without knowing the CI, cannot conclude the significance of the findings.

Methods -

Criteria considered for sample size calculation need to be mentioned.

Table 1:Since tumor size (based on the site) and thickness were considered in the eligibility criteria for the study, how can you justify having missing data about these variables?

Since tumour subtype was essentially important to the treatment (PDT), how can you justify of having missing data? Also, according to lines 136 and 137, BCC clasification was based on clinical examination. Hence, justify having missing values on tumor sub type.

Remove fullstops after mm (mm.)

Line 169 - ".. curette in which.." needs to be corrected as " ... curette by which.."

Results:

Line 252:Please provide the range of the CI, including the lower value, to get an informed decision by the reader, on the significance and inferiority.

Figure 2: Please provide axis titles

Discussion: need to provide more comparison with other studies, to have a rich discussion critically analyzing the global knowledge available

Reviewer #4: This manuscript presents data analysis from a investigator-initiated, single-blind, non-inferiority, randomized controlled, multicentre-study. The topic is of importance, the study was registered as a RCT (with a valid NCT number), and was approved by the respective IRB/Ethics Committee. While the study objectives sound interesting, is important, and on target, some shortcomings were observed, in regards to abiding by the CONSORT guidelines for conducting and reporting results of high-quality randomized controlled trials (RCTs). Some other (statistical) comments were also provided.

1. Methods:

Methods reporting need some work. An orderly manner is suggested, following CONSORT guidelines, without repeating information, such as Trial Design, Participant Eligibility Criteria and settings, Interventions, Outcomes, sample size/power considerations, Interim analysis and stopping rules, Randomization (details on random number generation, allocation concealment, implementation), Blinding issues, etc, should be mentioned. The authors are advised to create separate subsections for each of the possible topics (whichever necessary), and that way produce a very clear writeup. They are advised to write it carefully, following nice examples in the manuscript below:

https://www.sciencedirect.com/science/article/pii/S0889540619300010

Specific comments:

(a) For instance, the randomization and allocation concealment should be made very clear (they are NOT the same thing); the trial staff recruiting patients should NOT have the randomization list. Randomization should be prepared by the trial statistician, and he/she would not participate in the recruiting. The manuscript generates randomization via "computer-generated e-message". More details needed. Any reasoning, why a block randomization was not used, which is often recommended to ensure a balance in sample size across groups?

https://www.ncbi.nlm.nih.gov/pmc/articles/PMC2267325/

(b) Sample size/power: A sample size/power statement is made available, but its recommended to place it as a separate subsection within the Methods section, following CONSORT guidelines. Also, it is not clear what sample size formula was used to power the non-inferiority trial. Some relevant material might be here: https://www.ncbi.nlm.nih.gov/pmc/articles/PMC7808096/

(c) Statistical Analysis: Very straightforward, but with sketchy descriptions.

(c1) Not clear, what statistical test was used to test for non-inferiority.

(c2) Sample size was decently large; I was wondering why any formal regression analysis was also not performed.

(c3) Not clear, whether the associated test statistic in robust under possible non-Gaussian assumptions; however, with the large sample sizes, it might be working! Some clarifications would be helpful.

2. Results & Discussions/Conclusions:

(a) The authors should check that any statement of significance should be followed by a p-value in the entire Results section. Otherwise, the Results section look OK.

(b) Discussion section should state that the current findings are ONLY based on the random samples derived from this specific population, and should allude to future (larger) studies collected at other geographical areas to confirm the non-inferiority findings.

Reviewer #5: The authors are commended for conducting this trial in par with oncological concepts and research guidelines. In general, the conduct of the trail is scientifically sound. The following comments are given with the hope that these will be beneficial for the authors.

1. If there is a possibility, the reviewer prefers to request for a subgroup analysis of main outcomes of the participants with multiple lesions. Results of response of the lesions in same person would get affected by background factors. Therefore, subgroup analysis lesions in the same individual would increase the robustness of the findings.

2. Please provide more data on minimum distance of each close by lesions in the same person as there would be overlapping of treatment field.

3. Seems all the patients got treatment with photosensitizers target mitochondria rather than ER. Please confirm this. Otherwise needs a subgroup analysis for both types.

4. Please provide what measures were taken in minimization of bias in response assessment as It has been done by several individuals with their clinical experience.

5. Adverse effect profile need to be more descriptive.

6. If available please provide details on dose prescription of the beam with the duration of delivery, hemoglobin level and optimization prior to treatment etc, as these factors may potentially influence the response.

6. PLOS authors have the option to publish the peer review history of their article (what does this mean?). If published, this will include your full peer review and any attached files.

Reviewer #1: No

Reviewer #2: No

Reviewer #3: **Yes: **Ishanka Ayeshwari Talagala

Reviewer #4: No

Reviewer #5: No

---

## [Author Response · Author response to Decision Letter 0]

3 Jan 2024

Response to editor and reviewers

We thank you for reviewing our paper and greatly appreciate the effort and comments provided.

The original manuscript has been revised according to the response to the comments. Our answers to the each reviewer can be found under each comment and in italics. 

Changes in the the manuscript are shown in quotation marks. The line numbers are according to the clean manuscript.

Comments to the Author

1. Is the manuscript technically sound, and do the data support the conclusions?

Reviewer #1: Yes

Reviewer #2: Yes

Reviewer #3: Partly

Reviewer #4: Yes

Reviewer #5: Yes

Answer: More details on the statistical approach and the randomization process has been provided in the revised version of the manuscript.

2. Has the statistical analysis been performed appropriately and rigorously? 

Reviewer #1: Yes

Reviewer #2: I Don't Know

Reviewer #3: Yes

Reviewer #4: Yes

Reviewer #5: N/A

3. Have the authors made all data underlying the findings in their manuscript fully available?

Reviewer #1: No

Reviewer #2: No

Reviewer #3: Yes

Reviewer #4: Yes 

Reviewer #5: Yes

Answer: 

We recognize the benefit of being able to share data, but there are ethical restrictions on sharing raw data used this study according to requirements given by our Research Ethics Committee. 

The patients consented the data to be used for this particular study only, thus data cannot be used in other studies. By including data regarding study centers, anonymization may be breached since some treatment centers had a relative small number of included patients. 

However, a minimal data set, not including treatment centers may be requested. 

The contact information to which data requests may be sent is: kontakt@ikom.ntnu.no

4. Is the manuscript presented in an intelligible fashion and written in standard English?

Reviewer #1: Yes

Reviewer #2: Yes

Reviewer #3: Yes

Reviewer #4: Yes

Reviewer #5: Yes

5. Review Comments to the Author

Reviewer #1: 

A sub analysis between study sites can be included see any site specific variations, if any.

Answer: This was not the aim of the study and we have not taken such sub-analysis into account when calculation the number of tumours to be included. However, to accommodate the question, we have performed an analysis in which neither treatment center nor patient ID had any impact on the outcome (mixed model, p=0.7 for random effects). This is to be expected from the design, as we randomized individual lesions. We therefore did not pursue any study site sub analysis. 

Seems bit late in publishing as the the study was done 2012 -2014.

Answer: We would like to provide some background information as to why this study has taken so long to complete. This is an investigator-initiated study where funding for implementation of all aspects of the study was dependent on grants from non-commercial sources and the application processes associated with this has been very time-consuming. Inclusion of lesions (patients) was made from 2012 to 2014 with the last 3-year follow-up carried out in 2017. In 2018-19, the study-affiliated pathology departments were contacted for retrieval and sending of biopsy wax blocks to St. Olavs’s Hospital for preparation before assessment of histological BCC subtype and thickness could take place. The corona pandemic slowed down work from the beginning of 2020. Later, with a PhD candidate associated with the project, the final analysis of data and writing of the manuscript were completed.

Reviewer #2:

 - Mention the objectives of the study in the abstract as well.

Answer: The objectives of the studies are included in the material section of the revised manuscript.

- Some details of the design are not clearly stated, i.e. who generated the random allocation sequence, who enrolled participants, and who assigned participants to

Interventions.

Answer: More details are now added into the manuscript.

Patents were recruited from the dermatological centres participating in the study and were screen for study eligibility. Randomization was performed by a web-based randomization system developed and administered by the faculty of Medicine and Health Sciences, Norwegian University of Science (NTNU) and Technology, Trondheim, Norway. Tumours were randomized to receive the new or the standard treatment by use of the randomization module in WebCRF2, a program for data collection and randomisation administered by NTNU. Block randomization was used by treatment center. The block sizes were set by the system administrators and were set to vary. Both the order of block sizes and allocation sequence in each block were generated consecutively by the system. An administrator at NTNU initiated the randomization of each tumour in the program and the outcome was sent by email to the appointed study investigator who carried out the treatment. To ensure unpredictability of the random allocation in patients with multiple tumours, the investigators numbered each tumour according to a predetermined arrangement before the randomization took place. The numbering of tumours was recorded on the bodymap in the controlled report forms (CRFs).

Text from the “clean” manuscript, line 217 to 232:

“BCCs were randomised to receive the new or the standard treatment by use of a web-based system developed and administered by the Faculty of Medicine and Health Sciences, NTNU. Block randomisation was done by center where both the order of block sizes and allocation sequence of each block were generated consecutively by the system. An administrator initiated the randomisation system and the assignment was sent by email to the appointed study investigator who carried out the treatment.

To ensure unpredictability of the random allocation in patients with multiple BCCs, tumours were numbered consecutively, and recorded on the body map included in the case report forms (CRFs) before randomisation. The distance between BCCs had to be clinically ≥30 mm apart to be regarded as two individual tumours. The numbering started on the front side of the patient’s body and from top to bottom. If two tumours were located on the same horizontal line, the numbering first followed the tumour located furthest to the right side of the patient’s body. The corresponding system was then applied to the patient’s back. The first tumour was randomised to one of the two treatment regimens and the second was allocated to the other regimen. A third tumour was randomised to one of the two treatment regimens and a fourth allocated to the other regimen and so on.”

- For binary outcomes, presentation of both absolute and relative effect sizes is recommended (Table five)

Answer: We perceive the question as applying to Tables 3 and 4. We have considered reporting relative risk (RR) as well as risk differences. However, to avoid confusion we would prefer to not add more columns to Table 3 and 4. The interested reader may easily calculate the RR themselves from available data E.g., for the first row of Table 3, the RR for non-failures is 139/(139+61) / (146/(146+34) = 0.86 in “favour” of the new treatment. 

- Sources of potential Bias need to be addressed in the discussion section.

Answer: We have added information on this area in the discussion section. The study results cannot be generalized to all BCCs but apply to tumours that met the study eligibility criteria and from a fair-skinned population. 

Furthermore, we have added that reporting of conceivable adverse events may have been incomplete due to the patient’s varying ability to remember symptoms over a 3-month period. Also, that several dermatologists could be involved in the assessment of treatment results during the study period at the various centres, and that this may have led to a less uniform assessment of the findings. However, this practice is in accordance with common clinical practice and may thus increase the generalizability of results.

Text from the “clean” manuscript, line 374 to 383:

“However, the results do not apply to all BCC but to tumours that met the study eligibility criteria and from a fair-skinned population. Limitations also include that all BCCs were histologically verified before inclusion, which affects the generalizability of the results and the use of punch biopsies for histological examination since they only offer information from a small, selected area of the tumours[27]. Reporting of conceivable adverse events may have been incomplete due to the patient’s different ability to recall symptoms over a 3-month period. Different dermatologists may have been involved in the assessment of treatment areas at the various centres during the study period, and this may have led to a less uniform assessment of outcomes. However, the practise was carried out in accordance with common clinical practise and may thus increase the generalizability of the results.”

Reviewer #3: Congratulations to the authors for this important study.

My suggestions to further improve the manuscript are as follows:

Abstract -

Methods: sample size calculation details need to be given.

Answer: sample size calculation details is added in the abstract of the revised manuscript

Text from the “clean” manuscript, line 45 to 47: 

“With a non-inferiority margin of 0.1 and an expected probability complete response of 0.85, 190 tumours were required in each group.”

Prognostic factors assessed for cosmetic outcomes, not mentioned.

Answer: When reviewing the manuscript, we realize that the end sentence of the abstract may cause confusion. Prognostic factor refers to patient and tumour characteristics and not cosmetic outcome. The sentence has been altered and is hopefully now more comprehensible. 

Text from the “clean” manuscript, line 11 to 104.

“We also aimed to evaluate the cosmetic outcome and in addition explore prognostic factors such as the patient’s sex and age and tumour location, size, clinical, and histological subtypes, and thickness that may contribute to treatment failure in the groups.”

Results: The range of the 97.5% CI needs to be given for informed decision making.

Answer: To improve precision where it matters, a one-sided CI is commonly reported for non-inferiority studies as the lower end is not of interest. E.g., the 97.5 % CI for the difference reported in the first row of Table 4 is actually (-1.00 , 0.192) meaning that the new treatment is compatible with a lot better but also 19.2 % worse than the standard. The latter still extends beyond the 10 % non-inferiority margin. The traditional 95 % CI (not presented in the manuscript) is (0.03, 0.20) so the one-sided CI is about 1 absolute percent more precise then the two-sided.

Conclusion: Without knowing the CI, cannot conclude the significance of the findings.

Answer: See the above (previous) answer

Methods -

Criteria considered for sample size calculation need to be mentioned. 

Answer: The criteria for sample size calculation is given in a separate section under Material and Methods. A two-sided sample size calculation was used ,(significant level of 0.05 and power of 0.80). The final statistical analyses was performed with an one-sided test and by this, it is possible to present one CI (97.5%) only. This improves precision where it matters. 

A one-sided CI is commonly reported for non-inferiority studies as the lower end is not of interest. E.g., the 97.5 % CI for the difference reported in the first row of Table 4 is actually (-1.00 , 0.192) meaning that the new treatment is compatible with a lot better but also 19.2 % worse than the standard. The latter still extends beyond the 10 % non-inferiority margin. The traditional 95 % CI (not presented in the manuscript) is (0.03, 0.20) so the one-sided CI is about 1 absolute percent more precise then the two-sided.

Text from the “clean” manuscript, line 209 to 214.

“Sample size was determined by StatXact version 9.0 (Cytel Software Corp, Waltham, USA), based on anticipated complete response probability of 0.85 obtained from early publications [6, 7] and a non-inferiority margin of 10%; thus, aiming to demonstrate that the new regimen was not >0.1 inferior to the standard regimen. With a significance level at 0.05 and power at 0.80, each group required 190 tumours. Because multiple tumours were randomised within patients, no adjustments for patient identity were made.”

Text from the “clean” manuscript, line 242 to 244:

“The final analysis was performed by StatExact version 10.0 (Cytel Software Corp.). We used an exact non-inferiority test with a margin of 0.1 and corresponding one-sided 97.5% confidence intervals (CI) since one-sided CIs are customary in non-inferiority studies.”

Table 1: Since tumor size (based on the site) and thickness were considered in the eligibility criteria for the study, how can you justify having missing data about these variables?

Since tumour subtype was essentially important to the treatment (PDT), how can you justify of having missing data? Also, according to lines 136 and 137, BCC clasification was based on clinical examination. Hence, justify having missing values on tumor sub type.

Answer: Assessment of clinical tumour size, thickness and subtype was carried out by the investigators and, as pointed out, all these parameters are included in the eligibility criteria for this study. Therefore, the investigators had to be in possession of all these parameters to allow inclusion of tumours into the study, However, in a few cases, investigators have failed to record these parameters into the patient’s case report form (CRF) which forms the basis of the study data file. Data that are not recorded in the patient’s CRF will thus appear as missing in the file. 

Line 169 - ".. curette in which.." needs to be corrected as " ... curette by which.."

Answer: this has been corrected

Results:

Line 252:Please provide the range of the CI, including the lower value, to get an informed decision by the reader, on the significance and inferiority.

Answer: We hope this has been answered satisfactory earlier.

Figure 2: Please provide axis titles

Answer: “Cosmetic outcome” has been provided to the y-axis of the figure. 

Discussion: need to provide more comparison with other studies, to have a rich discussion critically analyzing the global knowledge available

Answer: We understand the question and lack of “equivalent” research was an inspiration to carry out this study. This is the first randomized controlled study comparing a simplified regimen to the standard two treatments of MAL-PDT in BCC. The practice of two treatment session at an interval of 1 week emerges on results from early, open-label studies which reported generally a higher effect than results from studies with one treatment. This has been addressed in the introduction of the manuscript and we do not wish to repeat this information in the discussion. The practice of two treatments has been recommended over many years without being appropriately tested, thus, to the best of our knowledge, there are no previous directly comparable studies available. This is now stated in the discussion section. However, we have included recent studies on alternative approaches to try optimizing the standard PDT regime in the discussion.

Text from the “clean” manuscript, line 354 to 357:

“The practise of two MAL-PDT sessions for BCC has been recommended for about two decades without being properly tested. To the best of our knowledge, this is the first randomised controlled study comparing a simplified regimen consisting of a single PDT session with the possibility of one re-treatment, with standard two treatments.”

Reviewer #4: This manuscript presents data analysis from a investigator-initiated, single-blind, non-inferiority, randomized controlled, multicentre-study. The topic is of importance, the study was registered as a RCT (with a valid NCT number), and was approved by the respective IRB/Ethics Committee. While the study objectives sound interesting, is important, and on target, some shortcomings were observed, in regards to abiding by the CONSORT guidelines for conducting and reporting results of high-quality randomized controlled trials (RCTs). Some other (statistical) comments were also provided.

1. Methods:

Methods reporting need some work. An orderly manner is suggested, following CONSORT guidelines, without repeating information, such as Trial Design, Participant Eligibility Criteria and settings, Interventions, Outcomes, sample size/power considerations, Interim analysis and stopping rules, Randomization (details on random number generation, allocation concealment, implementation), Blinding issues, etc, should be mentioned. The authors are advised to create separate subsections for each of the possible topics (whichever necessary), and that way produce a very clear writeup. They are advised to write it carefully, following nice examples in the manuscript below:

https://www.sciencedirect.com/science/article/pii/S0889540619300010

Answer: Additional subsections for specific topics have been included to the methods section of the revised manuscript.

Specific comments:

(a) For instance, the randomization and allocation concealment should be made very clear (they are NOT the same thing); the trial staff recruiting patients should NOT have the randomization list. Randomization should be prepared by the trial statistician, and he/she would not participate in the recruiting. The manuscript generates randomization via "computer-generated e-message". More details needed. Any reasoning, why a block randomization was not used, which is often recommended to ensure a balance in sample size across groups?

Answer (a): We have in the revised manuscript included more detailed information in the material section regarding the points made. Randomisation was performed by a web-based randomisation system developed and administered by the Faculty of Medicine and Health Sciences, Norwegian University of Science (NTNU) and Technology, Trondheim, Norway. Tumours were randomized to receive the new or the standard treatment by use of the randomization module in WebCRF2, a program for data collection and randomization administered by NTNU. Block randomisation was used by treatment center. The block sizes were set by the system administrators and were set to vary. Both the order of block sizes and allocation sequence in each block were generated consecutively by the system. An administrator at NTNU initiated the randomization of each tumour in the program and the outcome was sent by email to the appointed study investigator who carried out the treatment.

In patients with multiple tumours the first tumour was 

randomized to one of the two regimes and the second was allocated to the other regime. In the case of a third tumour, it was randomized to one of the two regimes and in case of a a fouth, this would be allocated to the other regime and so forth. To ensure unpredictability of the random allocation in patients with multiple tumors, each tumour was numbered according to a predetermined arrangement before the randomization took place and the numbering of tumours was recorded to the bodymap in the controlled report forms (CRFs). Different BCCs within each patient were numbered the following way: numbering started from the patient’s frontside of the body and from top to bottom. If two tumours were located on the same horizontal line, the numbering first followed the tumour located furthest to the right side of the patient’s body. When all tumours on the pasients frontside were numbered, the same system applied for numbering tumours on the patient’s back. 

https://www.ncbi.nlm.nih.gov/pmc/articles/PMC2267325/

Text from the “clean” manuscript, line 217 to 232:

“BCCs were randomised to receive the new or the standard treatment by use of a web-based system developed and administered by the Faculty of Medicine and Health Sciences, NTNU. Block randomisation was done by center where both the order of block sizes and allocation sequence of each block were generated consecutively by the system. An administrator initiated the randomisation system and the assignment was sent by email to the appointed study investigator who carried out the treatment.

To ensure unpredictability of the random allocation in patients with multiple BCCs, tumours were numbered consecutively, and recorded on the body map included in the case report forms (CRFs) before randomisation. The distance between BCCs had to be clinically ≥30 mm apart to be regarded as two individual tumours. The numbering started on the front side of the patient’s body and from top to bottom. If two tumours were located on the same horizontal line, the numbering first followed the tumour located furthest to the right side of the patient’s body. The corresponding system was then applied to the patient’s back. The first tumour was randomised to one of the two treatment regimens and the second was allocated to the other regimen. A third tumour was randomised to one of the two treatment regimens and a fourth allocated to the other regimen and so on.”

(b) Sample size/power: A sample size/power statement is made available, but its recommended to place it as a separate subsection within the Methods section, following CONSORT guidelines. Also, it is not clear what sample size formula was used to power the non-inferiority trial. Some relevant material might be here: https://www.ncbi.nlm.nih.gov/pmc/articles/PMC7808096/

Answer (b): Sample size/power is now placed under a separate subsection in the revised manuscript. We apologize for some confusion probably caused by a "copy-paste" error in relation to the description of sample size that was determined by Cytel Studio version 9.0 and not Stata. We have changed the mansucript text accordingly. Cytel studio employs an iterative algorithm for the calculation, thus a formula is not available"

(c) Statistical Analysis: Very straightforward, but with sketchy descriptions.

Answer (c): We have aimed to present the results in a clear and comprehensible manner. 

(c1) Not clear, what statistical test was used to test for non-inferiority.

Answer: This is known as "Unconditional test of non-inferiority using difference of two binomial proportions", with the exact option (i.e. an iterative procedure) by Cytel Studio. 

(c2) Sample size was decently large; I was wondering why any formal regression analysis was also not performed.

Answer: According to study design and endpoints, a regression analysis was not warranted. However, we checked for importance of centre and patient ID by entering these as random factors in mixed model but found no evidence of such effects (see also above).

(c3) Not clear, whether the associated test statistic in robust under possible non-Gaussian assumptions; however, with the large sample sizes, it might be working! Some clarifications would be helpful.

Answer: A difference between two binomial proportions may well be approximated by a Gaussian distribution with large sample sizes. In this case an iterative exact solution is employed, by Cytel Studio.

2. Results & Discussions/Conclusions:

(a) The authors should check that any statement of significance should be followed by a p-value in the entire Results section. Otherwise, the Results section look OK.

Answer: Additional p-values has been added in the results section. 

(b) Discussion section should state that the current findings are ONLY based on the random samples derived from this specific population, and should allude to future (larger) studies collected at other geographical areas to confirm the non-inferiority findings.

Answer: We agree and have in the revised manuscript added a sentence about limitation of generalization of results related to the study eligibility criteria and population. 

Text from the “clean” manuscript, line 374 to 375:

“However, the results do not apply to all BCC but to tumours that met the study eligibility criteria and from a fair-skinned population.” 

Reviewer #5: The authors are commended for conducting this trial in par with oncological concepts and research guidelines. In general, the conduct of the trail is scientifically sound. The following comments are given with the hope that these will be beneficial for the authors.

1. If there is a possibility, the reviewer prefers to request for a subgroup analysis of main outcomes of the participants with multiple lesions. Results of response of the lesions in same person would get affected by background factors. Therefore, subgroup analysis lesions in the same individual would increase the robustness of the findings.

Answer: We have avoided this possible bias by randomizing BCCs within each patient in cases where the patient had more than one tumour. This is described in the randomization section of the manuscript. 

Text from the “clean” manuscript, line 213 to 214:

“Because multiple tumours were randomised within patients, no adjustments for patient identity were made.”

2. Please provide more data on minimum distance of each close by lesions in the same person as there would be overlapping of treatment field.

Answer: The distance between BCCs in patients with multiple tumours had to be clinically ≥ 30 mm apart to be regarded as two individual tumours. This information has been added to the randomization section of the revised manuscript.

Text from the “clean” manuscript, line 225 to 226:

“The distance between BCCs had to be clinically ≥30 mm apart to be regarded as two individual tumours.”

3. Seems all the patients got treatment with photosensitizers target mitochondria rather than ER. Please confirm this. Otherwise needs a subgroup analysis for both types.

Answer: We can confirm that all tumours were treated with topical metylaminolevulinate which through intracellular formation of photoactive porphyrins together with oxygen and light primarily target mitochondria.

4. Please provide what measures were taken in minimization of bias in response assessment as It has been done by several individuals with their clinical experience.

Answer: We thank you for bringing this to our attention. The point is now addressed in the revised version of the manuscript. For valid assessment of the treatment results, dermatologists working at the various centres carried out this assignment. All the study centres had offered patients PDT of BCC over many years, thus the dermatologist were well acquainted with the method and with clinical assessment of results. Many dermatologists could be involved in assessment of the treatment outcomes during the study period, but one examiner assessed each individual outcome. No courses were conducted or brochures material prepared in advance for this specific assignment that could have contributed to a more uniform assessment. However, we believe that the assessments of the treatments results were carried out in accordance with common clinical practice and thus increase the generalizability of the results. 

Text from the “clean” manuscript 380 to 383:

“Different dermatologists may have been involved in the assessment of treatment areas at the various centres during the study period, and this may have led to a less uniform assessment of outcomes. However, the practise was carried out in accordance with common clinical practise and may thus increase the generalizability of the results.”

5. Adverse effect profile need to be more descriptive.

Answer: One serious adverse event was recorded during the study. One incident provides too limited of a basis to be able to present an adverse event profile in the manuscript. PDT for BCC has for two decades been approved for two treatments at an interval of one week. An increase of adverse events after use of a simplified regime (possibility of less treatment) was not expected. However, in accordance with good clinical practice, accounts were taken of the possible occurrence of any such events, Adverse events occurring in the period from treatment to the 3-month follow-up were recorded. More details regarding adverse events and serious adverse events have been added to the material section of the revised manuscript. 

Text from the “clean” manuscript line 200 to 206:

“Any adverse events (AEs) that occurred in the period from treatment to the 3-month follow-up were reported and described by their duration, severity, relationship to treatment and according to the need of other specific therapy. Serious adverse event (SAE) were to be reported according to specified procedures whether they were considered related to study treatment or not. Local reactions, such as erythema, pain, and weeping, were regarded as conceivable events and reported as number of days present. AEs could be reported spontaneously by the patient or through open (non-leading) questioning.”

6. If available please provide details on dose prescription of the beam with the duration of delivery, hemoglobin level and optimization prior to treatment etc, as these factors may potentially influence the response.

Answer: For this study we used the Aktilite CL128 lamp with light-emitting diodes (LEDs). The LED array is placed in a rectangular pattern and the lamp emits a light with a narrow spectrum at approximately 630 nm and a fluence rate of 70 x 100 mW/cm2. The light exposure is commonly 7-9 min giving a light dose of approximately 37 J/cm2. More details of the light source are added to the material section of the revised manuscript. All BCC’s were treated with topically applied metylaminolevuninate before light exposure. 

The international PDT guidelines have no recommendation on the use of the optimization of haemoglobin before PDT of skin cancer, nor is such a procedure common practice. As this was a clinical study, such a procedure was not performed. 

Text from the “clean” manuscript line 168 to 171:

“The cream was left for 3 h before being removed and the treatment area was exposed to light-emitting diodes (Aktilite®) with a peak wavelength of 630 nm, fluence rate of 70 x 100 mW/cm2 and exposure typically for 7-9 minutes giving a total light dose of about 37 J/cm2.”

6. PLOS authors have the option to publish the peer review history of their article (what does this mean?). If published, this will include your full peer review and any attached files.

Do you want your identity to be public for this peer review? For information about this choice, including consent withdrawal, please see our Privacy Policy.

Reviewer #1: No

Reviewer #2: No

Reviewer #3: Yes: Ishanka Ayeshwari Talagala

Reviewer #4: No

Reviewer #5: No

---

## [Decision Letter · Decision Letter 1]

6 Feb 2024

PONE-D-23-27160R1New, simplified versus standard photodynamic therapy (PDT) regimen for superficial and nodular basal cell carcinoma (BCC): A single blind, non-inferiority, randomised controlled multicentre study.PLOS ONE

Dear Dr. Christensen,

Thank you for submitting your manuscript to PLOS ONE. After careful consideration, we feel that it has merit but does not fully meet PLOS ONE’s publication criteria as it currently stands. Therefore, we invite you to submit a revised version of the manuscript that addresses the points raised during the review process.

**ACADEMIC EDITOR:**

The authors have addressed all comments of three reviewers successfully. Reviewer-3 has suggested two more revisions.

We look forward to receiving your revised manuscript.

Kind regards,

Pasyodun Koralage Buddhika Mahesh

Academic Editor

PLOS ONE

Journal Requirements:

Reviewers' comments:

Reviewer's Responses to Questions

**Comments to the Author**

1. If the authors have adequately addressed your comments raised in a previous round of review and you feel that this manuscript is now acceptable for publication, you may indicate that here to bypass the “Comments to the Author” section, enter your conflict of interest statement in the “Confidential to Editor” section, and submit your "Accept" recommendation.

Reviewer #1: (No Response)

Reviewer #3: All comments have been addressed

Reviewer #4: All comments have been addressed

Reviewer #5: All comments have been addressed

2. Is the manuscript technically sound, and do the data support the conclusions?

Reviewer #1: Yes

Reviewer #3: Yes

Reviewer #4: (No Response)

Reviewer #5: Yes

3. Has the statistical analysis been performed appropriately and rigorously? 

Reviewer #1: Yes

Reviewer #3: Yes

Reviewer #4: (No Response)

Reviewer #5: N/A

4. Have the authors made all data underlying the findings in their manuscript fully available?

Reviewer #1: Yes

Reviewer #3: Yes

Reviewer #4: (No Response)

Reviewer #5: Yes

5. Is the manuscript presented in an intelligible fashion and written in standard English?

Reviewer #1: Yes

Reviewer #3: Yes

Reviewer #4: (No Response)

Reviewer #5: Yes

6. Review Comments to the Author

Reviewer #1: (No Response)

Reviewer #3: Page 3 line 43: "...with objective to.." better to correct as "..with the objective of investigating.."

Table 4: per protocol analysis : the difference need to be corrected as 9.3%

Reviewer #4: (No Response)

Reviewer #5: (No Response)

7. PLOS authors have the option to publish the peer review history of their article (what does this mean?). If published, this will include your full peer review and any attached files.

Reviewer #1: No

Reviewer #3: No

Reviewer #4: No

Reviewer #5: No

---

## [Author Response · Author response to Decision Letter 1]

7 Feb 2024

Response to editor and reviewer

We thank you for reviewing our paper and greatly appreciate the effort and comments provided.

The original manuscript has been revised according to the response to the comments. 

Our answers to can be found under the comments and in italic. 

Comments from reviewer #3: 

Page 3 line 43: "...with objective to.." better to correct as "..with the objective of investigating.."

Table 4: per protocol analysis: the difference need to be corrected as 9.3%

Answer: 

The wording on page 3, line 43 has been changed to “…with the objective of investigating"

The number in Table 4 has been corrected from 9.4 % to 9.3%.

---

## [Editor Report · Decision Letter 2]

15 Feb 2024

New, simplified versus standard photodynamic therapy (PDT) regimen for superficial and nodular basal cell carcinoma (BCC): A single blind, non-inferiority, randomised controlled multicentre study.

PONE-D-23-27160R2

Dear Dr. Christensen,

We’re pleased to inform you that your manuscript has been judged scientifically suitable for publication and will be formally accepted for publication once it meets all outstanding technical requirements.

Kind regards,

Pasyodun Koralage Buddhika Mahesh

Academic Editor

PLOS ONE

Additional Editor Comments (optional):

The first "full stop" in line 241 (i.e. centres. blinded) should be removed. 
---

## [Editor Report · Acceptance letter]

26 Feb 2024

PONE-D-23-27160R2 

PLOS ONE

Dear Dr. Christensen, 

I'm pleased to inform you that your manuscript has been deemed suitable for publication in PLOS ONE. Congratulations! Your manuscript is now being handed over to our production team.

Kind regards, 

on behalf of

Dr. Pasyodun Koralage Buddhika Mahesh 

Academic Editor

PLOS ONE